# ILVS²Net: Illumination-Driven Non-Local Visual State Space Unfolding Network for Low-Light Enhancement

## Abstract

In low-light image enhancement (LLIE), deep unfolding methods have achieved remarkable success by bridging physical models with learnable modules. However, existing approaches often overlook the structured sparsity of illumination, which leads to oversmoothing and unstable recovery. To address this, we propose ILVS²Net, a deep Retinex unfolding network that explicitly integrates a group-sparse prior into each iteration. Specifically, we design two learnable proximal operator networks: a Non-Local Visual State Space (NLVSS) module that translates the grouping and shrinkage principle of group sparsity into a neural operator, effectively capturing long-range structural dependencies; and an Illumination Smoothing Operator (ISP) that enforces edge-preserving piecewise smoothness for coherent illumination estimation. By embedding these proximal operator networks into the unfolding process, our model achieves a stable closed-form update while dynamically adapting to complex illumination variations. Extensive experiments on five public benchmarks demonstrate that ILVS²Net consistently outperforms state-of-the-art methods in both quantitative metrics and perceptual quality. The code and pretrained models will be released.

## 1 Introduction

Low-light images, captured under challenging lighting conditions, often exhibit color distortion, detail loss, and extremely low contrast. These issues not only reduce human visual perception but also degrade the performance of downstream computer vision tasks, such as semantic segmentation Li et al. (2022); Hou et al. (2024), object detection He et al. (2023), and autonomous driving. As a result, low-light image enhancement has gained significant attention.

Current LLIE methods are generally categorized into three approaches: Traditional, Deep Learning-Based, and Deep Unfolding-Based methods. Traditional methods, such as histogram equalization Hummel (1975); Arici et al. (2009); Pizer et al. (1987); Abdullah-Al-Wadud et al. (2007) and gamma correction Huang et al. (2013); Wang et al. (2019b), often fail to maintain a natural appearance under complex lighting. Retinex theory Land (1977), inspired by the Human Visual System (HVS), decomposes an image into reflectance and illumination components. While Deep Learning-Based methods have surpassed traditional techniques, they often rely on black-box structures that lack

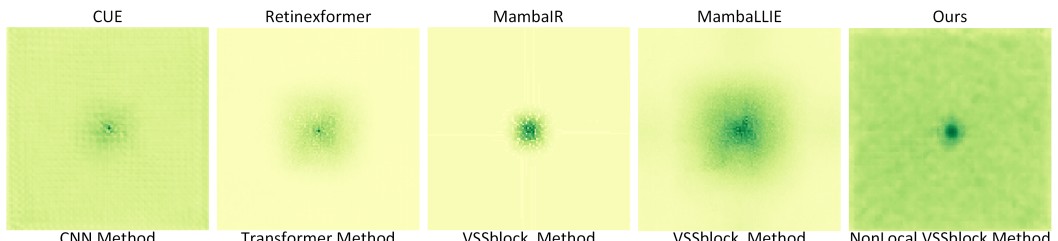

Figure 1: The Effective Receptive Field (ERF) visualization for CUE Zheng et al. (2023a), Retinexformer Cai et al. (2023), MambaIRGuo et al. (2024), MambaLLIE Weng et al. (2024) and the proposed ILVS²Net. A larger ERF is indicated by a more extensively distributed dark area.

interpretability and require numerous learnable parameters Wang et al. (2019a); Zhang et al. (2019); Chen et al. (2018); He et al. (2023); Zhang et al. (2021b). Deep Unfolding-Based LLIE methods have emerged with Retinex theory and achieved great success by incorporating physical priors into network architecture Wu et al. (2022a); Zhou et al. (2023b); Zheng et al. (2023a).

However, most Retinex-based deep unfolding methods treat Retinex components equally, ignoring their independent characteristics. The challenge of balancing illumination and reflectance during decomposition often results in illumination components that are oversmoothed and inconsistent. This inevitably leads to overexposure or underexposure in the enhanced images. Illumination is not merely another image feature, but a light-driven signal characterized by: (1) Non-local spatial correlations, as illumination at one point often depends on distant light sources and scene geometry (Kajiya, 1986); and (2) Piecewise-smooth variations, where intensity changes gradually within homogeneous regions but can jump sharply at object boundaries (Rudin et al., 1992). By neglecting these priors, existing architectures misinterpret lighting gradients as textures or noise Bai et al. (2024); Zhang et al. (2024b); Cai et al. (2023); Weng et al. (2024). A model's receptive field must accommodate both local and global contexts to handle the natural variations of illumination across different spatial scales. As shown in Fig. 1, existing methods suffer from limited receptive fields, capturing only local or only global information, but rarely both, which often leads to oversmoothing.

To address these challenges, we revisit Retinex decomposition through the lens of structured group sparsity. Illumination can be modeled as belonging to non-local groups of structurally similar patches, where a group-sparse prior enforces two properties simultaneously: (i) redundancy reduction across groups, and (ii) structural preservation within each group. This perspective directly inspires the design of our network, where the structured prior is translated into learnable proximal operator networks. We therefore propose two novel modules: (1) the Non-Local Visual State Space (NLVSS), which serves as a neural approximation of the group-sparse proximal operator, capturing long-range dependencies and enforcing non-local consistency ; and (2) the Illumination Smoothing Operator (ISP), which enforces piecewise-smoothness to ensure stable and edge-preserving illumination. As shown in Fig. 1, our approach exhibits diversity and dynamism in long-range modeling, explicitly reflecting its ability to preserve informative illumination variations.

The main contributions of this work are summarized as follows:

(1) We introduce the Non-local Visual State Space (NLVSS), a learnable proximal operator that embodies the group-sparse prior and captures non-local dependencies for illumination modeling.

(2) We propose an Illumination Smoothing Operator (ISP) that dynamically adjusts and reweights illumination estimation, enforcing piecewise-smoothness while preserving structural edges.

(3) We formulate a deep-unfolding framework that integrates NLVSS and ISP into each iteration, yielding a stable closed-form optimization solution with explicit physical priors.

(4) Extensive experiments show that our method achieves state-of-the-art performance in both quantitative metrics and visual quality, while also improving efficiency in downstream vision tasks.

## 2 RELATED WORK

### 2.1 TRADITIONAL LLIE METHODS

Traditional low-light image enhancement (LLIE) methods are generally categorized into three main types: histogram equalizationHummel (1975); Arici et al. (2009); Pizer et al. (1987); Abdullah-Al-Wadud et al. (2007), gamma correction Huang et al. (2013); Wang et al. (2019b), and Retinex-based approaches. However, under extreme conditions, these methods may introduce additional noise, leading to unnatural visual artifacts and a loss of fine details.

### 2.2 DEEP LEARNING-BASED LLIE METHODS

With the continuous advancements in deep learning, LLIE methods have gradually evolved to incorporate CNNs and Transformers. CNN-based approaches Wang et al. (2019a); Zhang et al. (2019); Chen et al. (2018) effectively learn spatial features; notably, Wei et al. Chen et al. (2018) pioneered an end-to-end Retinex decomposition. However, CNNs still face challenges in capturing

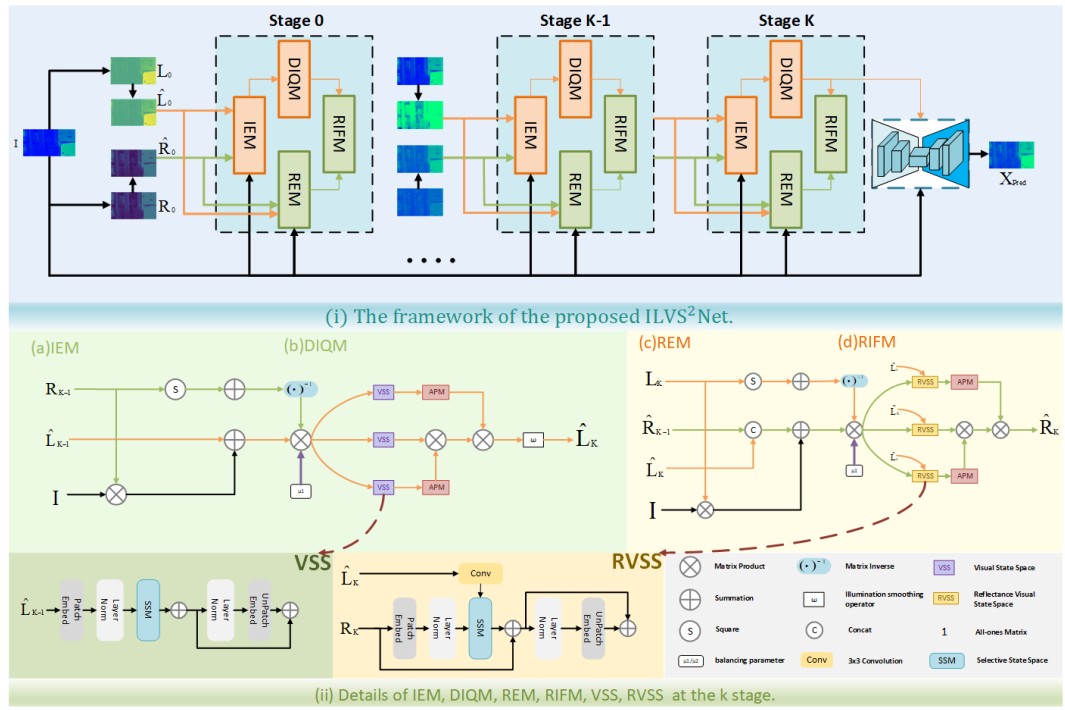

Figure 2: Framework of our proposed ILVS²Net. Showcasing the multi-stage unfolding of illumination ($L$) and reflectance ($R$) through IEM, DIQM, REM, RIFM, RVSS, and NLVSS blocks, culminating in the final enhanced image $X_{\text{pred}}$.

global context and long-range dependencies. To address these issues, Transformer models He et al. (2023); Zhang et al. (2021b) leverage attention mechanisms, with Star Zhang et al. (2021b) being one of the earliest Transformers applied to LLIE. Despite their impressive performance, Transformers demand high computational resources and large-scale training data. Recently, state space models (SSMs) like Mamba Gu & Dao (2023) have drawn attention for their memory efficiency and ability to handle long-range dependencies. While Mamba-based structures Zhang et al. (2024b); Bai et al. (2024) offer improvements to encoder-decoder designs for LLIE, they still struggle to effectively integrate both global and local information.

## 2.3 DEEP UNFOLDING-BASED LLIE METHODS

Deep unfolding-based LLIE methods integrate the interpretability of traditional model-driven approaches with the feature extraction capabilities of deep learning Wu et al. (2022a); Zhou et al. (2023b); Zheng et al. (2023a). Wu et al. Wu et al. (2022a) first introduced this idea by incorporating auxiliary variables and iterative ADMM updates to control reflectance and illumination. Subsequently, Zheng et al. Zheng et al. (2023a) added a learnable prior for illumination. Despite these advances, achieving smooth and consistent illumination that accurately captures both global and local features remains a challenging task.

## 3 METHOD

### 3.1 MODEL FORMULATION

To improve low-light image quality, we adopt Retinex theory (Land, 1977) and decompose a low-light image into illumination ($\mathbf{L}$) and reflectance ($\mathbf{R}$), optimizing each component:

$$\mathbf{L}, \mathbf{R} = \arg\min_{\mathbf{L}, \mathbf{R}} \|\mathbf{I} - \mathbf{R} \odot \mathbf{L}\|_F^2 + \beta\,\Theta(\mathbf{L}) + \delta\,\Omega(\mathbf{R}), \tag{1}$$

where $\Theta(\mathbf{L})$ and $\Omega(\mathbf{R})$ are regularization terms for $\mathbf{L}$ and $\mathbf{R}$, and $\beta, \delta > 0$ are balancing parameters.

**Definition 1 (Non-local Modeling Representation).** *Unlike methods such as RetinexMamba Bai et al. (2024) and RetinexFormer Cai et al. (2023), which extract patch-level features directly, we explicitly model non-local patch dependencies within Retinex components. Direct quadratic penalties on illumination and reflectance are often biased toward the degraded measurement* $\mathbf{I}$ *in Eq. 1, which can cause oversmoothing across unfolding iterations. Motivated by the observation that low-light images often contain regions with richer structural information, we introduce a group-sparse representation to capture such structured variations. Inspired by Zha et al. (2022), we integrate this representation into our Retinex model as follows:*

$$\hat{\mathbf{L}}, \hat{\mathbf{R}} = \arg\min_{\hat{\mathbf{L}}, \hat{\mathbf{R}}} \left\| \mathbf{X}_i^* - \hat{\mathbf{R}} \odot \hat{\mathbf{L}} \right\|_F^2 + \alpha\, \Phi(\hat{\mathbf{L}}) + \gamma\, \Psi(\hat{\mathbf{R}}), \tag{2}$$

where $\mathbf{X}_i^*$ denotes grouped data matrices, and $\hat{\mathbf{L}}, \hat{\mathbf{R}}$ are group-level Retinex components.

Following the tractability analysis in Bhan et al. (2013), we estimate $\hat{\mathbf{L}}$ and $\hat{\mathbf{R}}$ from $\mathbf{L}$ and $\mathbf{R}$ under a given group structure $\mathbb{G}$ by solving:

$$\Delta\hat{\mathbf{L}} = \mathbf{L} - \hat{\mathbf{L}}, \quad \text{s.t.} \quad \|\hat{\mathbf{L}}\|_{\mathbb{G},1,p} \le \lambda, \tag{3}$$

$$\Delta\hat{\mathbf{R}} = \mathbf{R} - \hat{\mathbf{R}}, \quad \text{s.t.} \quad \|\hat{\mathbf{R}}\|_{\mathbb{G},1,p} \le \lambda, \tag{4}$$

where $\lambda > 0$ balances approximation accuracy and group sparsity, and $\|\cdot\|_{\mathbb{G},1,p} := \sum_{g\in\mathbb{G}} \|(\cdot)_{G_g}\|_p$ denotes the group-$\ell_{1,p}$ norm over non-overlapping non-local groups $\{G_g\}$.

To make optimization tractable, we relax the hard constraints in Eqs. 3–4 into penalized (Lagrangian) forms and define unified non-local group-sparsity regularizers:

$$\mathcal{R}_{\text{GroupSparse}}^{(\hat{L})}(\hat{\mathbf{L}}) = \sum_{g\in\mathbb{G}_L} \rho\big(\|\mathbf{P}_g^{(\hat{L})}\hat{\mathbf{L}}\|_2\big), \qquad \mathcal{R}_{\text{GroupSparse}}^{(\hat{R})}(\hat{\mathbf{R}}) = \sum_{g\in\mathbb{G}_R} \rho\big(\|\mathbf{P}_g^{(\hat{R})}\hat{\mathbf{R}}\|_2\big),$$

where $\mathbb{G}_L, \mathbb{G}_R$ index non-local groups, $\mathbf{P}_g^{(\cdot)}$ extracts the $g$-th group (e.g., a stack of similar patches/channels), and $\rho(\cdot)$ is a sparsity-promoting potential. In practice, we use $\rho(t) = t$ (group-$\ell_{1,2}$ norm). These penalties are enforced via proximal mappings at each iteration of the unrolled network.

For the regularization term in Eq. 2, $\Phi(\hat{\mathbf{L}})$ is typically defined as $\|\hat{\mathbf{L}}\|_F^2$. In our formulation, however, we introduce a proximal operator network, denoted as $\varpi$, which adaptively refines $\hat{\mathbf{L}}$ to yield smoother illumination. Here, $\Phi, \Psi$ impose generic priors (e.g., smoothness), while $\mathcal{R}_{\text{GroupSparse}}$ enforces non-local structural consistency. By jointly leveraging these complementary properties, our model achieves improved accuracy and robustness.

Accordingly, the final form of the proposed model is:

$$\min_{\mathbf{L},\mathbf{R},\hat{\mathbf{L}},\hat{\mathbf{R}}} \|\mathbf{I} - \mathbf{R}*\mathbf{L}\|_F^2 + \mu_1\|\mathbf{L} - \hat{\mathbf{L}}\|_F^2 + \mu_2\|\mathbf{R} - \hat{\mathbf{R}}\|_F^2 + \alpha\Phi(\hat{\mathbf{L}}) + \gamma\Psi(\hat{\mathbf{R}})$$
$$+ \lambda_1 \mathcal{R}_{\text{GroupSparse}}^{(\hat{L})}(\hat{\mathbf{L}}) + \lambda_2 \mathcal{R}_{\text{GroupSparse}}^{(\hat{R})}(\hat{\mathbf{R}}) \tag{5}$$

where $\gamma, \alpha, \mu_1, \mu_2, \lambda_1, \lambda_2$ are regularization weights.

## 3.2 MODEL OPTIMIZATION

The algorithm alternates between updating the $\mathbf{L}, \mathbf{R}, \hat{\mathbf{L}}_k$ and $\hat{\mathbf{R}}_k$. Below we detail the $k$-th iteration of the optimization process.

**Updating $\mathbf{L}_k$.** Given $\mathbf{R}_{k-1}$ and $\hat{\mathbf{L}}_{k-1}$, we solve

$$\mathbf{L}_k = \arg\min_{\mathbf{L}} \|\mathbf{I} - \mathbf{R}_{k-1}*\mathbf{L}\|_F^2 + \mu_1\|\mathbf{L} - \hat{\mathbf{L}}_{k-1}\|_F^2, \tag{6}$$

This subproblem is a standard least squares formulation. Differentiating with respect to $\mathbf{L}$ and setting the derivative to zero yields the closed-form solution: $\mathbf{L_k} = \mathbf{Q}^{-1}\left[\mu_1\hat{\mathbf{L}}_{\mathbf{k-1}} + \mathbf{I}*\mathbf{R_{k-1}}\right]$, where $\mathbf{Q} = \mathbf{R_{k-1}}*\mathbf{R_{k-1}} + \mu_1\mathbf{1}$, $\mathbf{1}$ is an all-ones matrix, and $\mu_1$ is a balancing parameter.

**Updating $\hat{\mathbf{L}}_k$.** Given $\mathbf{L}_k$, we refine it by solving

$$\hat{\mathbf{L}}_k = \arg\min_{\hat{\mathbf{L}}} \mu_1\|\mathbf{L}_k - \hat{\mathbf{L}}\|_F^2 + \alpha\, \Phi(\hat{\mathbf{L}}) + \lambda_1 \mathcal{R}_{\text{GroupSparse}}^{(\hat{L})}(\hat{\mathbf{L}}), \tag{7}$$

where $\Phi$ and $\mathcal{R}_{\text{GroupSparse}}^{(\hat{L})}$ denote illumination priors. Since a closed-form solution is intractable, we approximate the proximal mappings using two learned networks:

$$\mathbf{U}_k = \mathcal{P}_\theta^{(\text{GS})}(\mathbf{L}_k), \quad \hat{\mathbf{L}}_k = \varpi(\mathbf{U}_k). \tag{8}$$

**Updating $\mathbf{R}_k$.** Given $\mathbf{L}_k$ and $\hat{\mathbf{R}}_{k-1}$, the reflectance is updated via

$$\mathbf{R}_k = \arg\min_{\mathbf{R}} \|\mathbf{I} - \mathbf{R}*\mathbf{L}_k\|_F^2 + \mu_2\|\mathbf{R} - \hat{\mathbf{R}}_{k-1}\|_F^2, \tag{9}$$

which leads to the closed-form update $\mathbf{R_k} = \mathbf{D}^{-1}\left[\mu_2\hat{\mathbf{R}}_{\mathbf{k-1}} + \mathbf{I} * \mathbf{L_k}\right]$, where $\mathbf{D} = \mathbf{L_k} * \mathbf{L_k} + \mu_2\mathbf{1}$, and $\mu_2$ is a balancing parameter.

**Updating $\hat{\mathbf{R}}_k$.** Finally, given $\mathbf{R}_k$, we solve

$$\hat{\mathbf{R}}_k = \arg\min_{\hat{\mathbf{R}}} \mu_2\|\mathbf{R}_k - \hat{\mathbf{R}}\|_F^2 + \gamma\,\Psi(\hat{\mathbf{R}}) + \lambda_2\mathcal{R}_{\text{GroupSparse}}^{(\hat{R})}(\hat{\mathbf{R}}). \tag{10}$$

Since reflectance primarily encodes scene details and textures, it does not require the additional piecewise-smooth prior enforced for illumination, so a single proximal operator suffices. Moreover, according to Retinex theory, reflectance restoration should be conditioned on illumination. Hence, we concatenate illumination and reflectance as inputs to the proximal operator network:

$$\hat{\mathbf{R}}_k = \mathcal{P}_\theta^{(R)}(\hat{\mathbf{L}}_k,\ \mathbf{R}_k). \tag{11}$$

### 3.3 Deep Network Architecture

Rather than treating iterations as an independent algorithm, we design a deep network with $N$ stages, each mimicking an optimization-inspired iteration. As shown in Figure 2, ILVS$^2$Net alternates among the Illumination Estimation Module (IEM), Reflection Estimation Module (REM), Dynamic Illumination Quantification Module (DIQM), and Reflection–Illumination Fusion Module (RIFM). The illumination $\mathbf{L}$ is initialized to the maximum pixel value of the input, while the reflectance $\mathbf{R}$ is obtained by pixel-wise division, with $\hat{\mathbf{L}}$ and $\hat{\mathbf{R}}$ set accordingly. In the final stage, the enhanced image is reconstructed as $\mathbf{X}_{\text{pred}} = \text{RecModule}(\mathbf{I}, \hat{\mathbf{L}}_k)$. The detailed architecture and algorithmic design are described below.

**Structure of the IEM.** The Illumination Estimation Module (IEM) is designed based on Eq. 6, where $\text{Module}_\mathcal{X}$ implements the update of $\mathbf{L}_k$:

$$\mathbf{L_k} = \text{Module}_\mathcal{X}(\mathbf{I}, \mathbf{R_{k-1}}, \hat{\mathbf{L}}_{\mathbf{k-1}}, \mu_1) = \mathbf{Q}^{-1}\left[\mu_1\hat{\mathbf{L}}_{\mathbf{k-1}} + \mathbf{I} * \mathbf{R_{k-1}}\right], \tag{12}$$

which follows the same closed-form update but with all parameters made learnable to enhance stability and generalization.

**Structure of the DIQM.** According to Eq. 7, to ensure the smoothness and continuity of illumination, we employ two learnable proximal operator networks. The first corresponds to the *group-sparse prior*, translated into a learnable operator. Specifically, the non-local group-sparse penalty is

$$\mathcal{R}_{\text{GroupSparse}}^{(\hat{L})}(\hat{\mathbf{L}}) = \sum_{g \in \mathbb{G}_L} \rho(\|\mathbf{P}_g^{(\hat{L})}\hat{\mathbf{L}}\|_2),$$

where the grouping operator $\mathbf{P}_g$ aggregates mutually similar patches or channels. This step mirrors non-local operations, enabling retrieval and aggregation of repeated structures across the image. The within-group shrinkage is implemented by block $\ell_{1,2}$ penalties, which we realize through Visual State Space (VSS) dynamics that propagate structured dependencies and suppress noise.

**NLVSS architecture.** As shown in Fig. 2 and Fig. 5, we replace the convolutional layers of the Non-local network Wang et al. (2018) with VSS structures and an APM module. This design allows $\mathbf{L}$ to propagate both globally and locally. Two VSS layers yield $\mathbf{L}_{p_1}$ and $\mathbf{L}_{p_3}$, while the APM employs four global average-pooling layers at multiple scales. Another VSS layer extracts $\mathbf{L}_{p_2}$. Together, these components capture long-range and contextual illumination dependencies.

Table 1: Results on the LLIE task. The best two results are in **red** and **blue** fonts, respectively.

| Methods | Sources | Efficiency | | LOL-v1 | | | | LOL-v2-real | | | | LOL-v2-synthetic | | | |
|---|---|---|---|---|---|---|---|---|---|---|---|---|---|---|---|
| | | Para. ↓ | FLOPs ↓ | PSNR ↑ | SSIM ↑ | FID ↓ | BIQE ↓ | PSNR ↑ | SSIM ↑ | FID ↓ | BIQE ↓ | PSNR ↑ | SSIM ↑ | FID ↓ | BIQE ↓ |
| URetinex (Wu et al., 2022b) | CVPR22 | 0.36 | 233.09 | 21.33 | 0.835 | 85.59 | 30.37 | 20.44 | 0.806 | 76.74 | 28.85 | 24.73 | 0.897 | 33.25 | 33.46 |
| UFormer (Wang et al., 2022) | CVPR22 | 5.20 | 10.68 | 16.36 | 0.771 | 166.69 | 41.06 | 18.82 | 0.771 | 164.41 | 40.36 | 19.66 | 0.871 | 58.69 | 39.75 |
| Restormer (Zamir et al., 2022a) | CVPR22 | 26.13 | 144.25 | 22.43 | 0.823 | 78.75 | 33.18 | 19.94 | 0.827 | 114.35 | 37.27 | 21.41 | 0.830 | 46.89 | 35.06 |
| SNR-Net (Xu et al., 2022) | CVPR22 | 4.01 | 26.35 | 24.61 | 0.842 | 66.47 | 28.73 | 21.48 | 0.849 | 68.56 | 28.83 | 24.14 | 0.928 | 30.52 | 33.47 |
| SMG (Xu et al., 2023) | CVPR23 | 14.02 | 17.55 | 24.82 | 0.838 | 69.47 | 30.15 | 22.62 | 0.857 | 71.76 | 30.32 | 25.62 | 0.905 | 23.36 | 29.35 |
| Diff-Retinex (Yi et al., 2023) | ICCV23 | 56.88 | 198.16 | 21.98 | 0.852 | 51.33 | 19.62 | 20.17 | 0.826 | 46.67 | 24.18 | 24.30 | 0.921 | 28.74 | 26.35 |
| MRQ (Liu et al., 2023) | ICCV23 | 8.45 | 20.66 | 25.24 | 0.855 | 53.32 | 22.73 | 22.37 | 0.854 | 68.89 | 33.61 | 25.54 | 0.940 | 20.86 | 25.09 |
| IAGC (Wang et al., 2023c) | ICCV23 | — | — | 24.53 | 0.842 | 59.73 | 25.50 | 22.20 | 0.863 | 70.34 | 31.70 | 25.58 | 0.941 | 21.38 | 30.32 |
| DiffIR (Xia et al., 2023) | ICCV23 | 27.80 | 35.32 | 23.15 | 0.828 | 70.13 | 26.38 | 21.15 | 0.816 | 72.33 | 29.15 | 24.76 | 0.921 | 28.87 | 27.74 |
| CUE (Zheng et al., 2023b) | ICCV23 | 0.25 | 157.32 | 21.86 | 0.841 | 69.83 | 27.15 | 21.19 | 0.829 | 67.05 | 28.83 | 24.41 | 0.917 | 31.33 | 33.83 |
| GSAD (Jinhui et al., 2023) | NIPS23 | 17.17 | 670.33 | 23.23 | 0.852 | 51.64 | 19.96 | 20.19 | 0.847 | 46.77 | 28.85 | 24.22 | 0.927 | 19.24 | 25.76 |
| AST (Zhou et al., 2024) | CVPR24 | 19.90 | 13.25 | 21.09 | 0.858 | 87.67 | 21.23 | 21.68 | 0.856 | 91.81 | 25.17 | 22.25 | 0.927 | 37.19 | 28.78 |
| RetiMamba Bai et al. (2024) | ArXiv | 3.59 | 37.98 | 24.03 | 0.827 | 75.33 | 16.28 | 22.45 | 0.844 | 56.96 | 21.76 | 25.89 | 0.934 | 20.17 | 16.29 |
| MambaIR (Guo et al., 2024) | ECCV24 | 4.30 | 60.66 | 22.23 | 0.863 | 63.39 | 20.17 | 21.15 | 0.857 | 56.09 | 24.46 | 25.75 | 0.937 | 19.75 | 20.37 |
| Mamballie Weng et al. (2024) | NIPS24 | 2.28 | 20.85 | 23.24 | 0.861 | — | — | 22.95 | 0.847 | — | — | 25.87 | 0.940 | — | — |
| CIDNet Yan et al. (2025) | CVPR25 | 1.88 | 7.57 | 23.50 | 0.900 | 46.69 | 14.77 | 24.11 | 0.871 | 48.04 | 18.45 | 25.71 | 0.942 | 18.60 | 15.87 |
| **ILVS²Net** | Ours | 3.42 | 11.27 | 24.33 | 0.910 | 42.89 | 13.46 | 23.06 | 0.888 | 36.44 | 18.35 | 26.16 | 0.960 | 18.27 | 15.03 |

| Input | URetinex | CUE | RetiMamba | MambaLLIE | CIDNet | Ours | Ground Truth |

Figure 3: Visual results on the low-light image enhancement task.

**Illumination smoothing operator.** Group sparsity alone may yield blocky or discontinuous illumination. In the unrolled optimization framework, this step corresponds to the proximal mapping of a piecewise-smooth prior, which is intractable to compute in closed form. Unlike a generic CNN filter, the ISP is not a post-hoc refinement block but the learnable realization of this proximal operator. Classical choices such as TV or WLS can be viewed as fixed forms of this mapping: they enforce smoothness but often lead to over-smoothing due to their hand-crafted nature. Our ISP generalizes these operators by adopting a data-adaptive design that combines VSS and IlluNet: VSS provides selective 2D scanning to capture non-local structural consistency, while IlluNet (five $5 \times 5$ convolutional layers with LeakyReLU) performs lightweight, edge-aware diffusion to enforce local smoothness. Thus, ISP reduces to TV/WLS under fixed linear parameters, but as a learnable proximal operator, it adaptively balances global consistency and local smoothness, ensuring both interpretability and stability within the optimization framework, beyond what a stand-alone CNN can offer. The final module for computing $\hat{\mathbf{L}}_k$ is defined as

$$\hat{\mathbf{L}}_k = \text{Module}_{\mathcal{F}}\big[\mathbf{L_k}; \boldsymbol{\Lambda}_{NLVSS}, \boldsymbol{\Lambda}_{\varpi}\big], \tag{13}$$

where $\boldsymbol{\Lambda}_{NLVSS}$ and $\boldsymbol{\Lambda}_{\varpi}$ denote the learnable parameters of the two proximal operator networks.

**Structure of the REM.** The Reflectance Estimation Module (REM) is derived from Eq. 9, with fixed parameters replaced by learnable ones:

$$\mathbf{R_k} = \text{Module}_{\mathcal{R}}\big(\mathbf{I}, \mathbf{L_{k-1}}, \hat{\mathbf{R}_{k-1}}, \mu_2\big) = \mathbf{D}^{-1}\left[\mu_2\hat{\mathbf{R}_{k-1}} + \mathbf{I} * \mathbf{L_{k-1}}\right]. \tag{14}$$

**Structure of the RIFM.** According to Eq. 10, to effectively fuse reflectance and illumination, we concatenate $\mathbf{L_k}$ and $\mathbf{R_k}$ along the channel dimension, and feed them into a proximal operator network to estimate $\hat{\mathbf{R}_k}$. This network adopts the NLVSS structure:

$$\hat{\mathbf{R}}_k = \text{Module}_{\mathcal{M}}\big[\mathbf{R_k}, \hat{\mathbf{L}}_k; \theta_{NLVSS}\big], \tag{15}$$

where $\theta_{NLVSS}$ denotes the learnable parameters. Finally, $\hat{\mathbf{L}}_k$ is concatenated with $\mathbf{I}$ and passed through a U-Net–style VSSBlock Weng et al. (2024) to reconstruct the enhanced image $\mathbf{X}_{\text{pred}}$.

## 4 EXPERIMENTS

**Implementation**: Experiments were conducted with PyTorch on NVIDIA GTX4090 GPUs, using a batch size of 8. We trained with the Adam optimizer Kingma & Ba (2014) ($\beta_1 = 0.9$, $\beta_2 = 0.99$) for 3000k iterations, starting with a learning rate of $2 \times 10^{-4}$ and halving it every 50k iterations. Hyperparameters $\mu_1$ and $\mu_2$ were initially set to 0.1 and increased by 0.05 at each stage. We employ a combination of mean absolute error (MAE) and a perceptual loss as our loss function(Cai et al., 2023).

Table 2: Results on the UIE task.

| Methods | Sources | *UIEB* | | | |
|---|---|---|---|---|---|
| | | PSNR ↑ | SSIM ↑ | UCIQE ↑ | UIQM ↑ |
| S-uwnet (Naik et al., 2021) | AAAI21 | 18.28 | 0.855 | 0.544 | 2.942 |
| PUIE (Fu et al., 2022) | ECCV22 | 21.38 | 0.882 | 0.566 | 3.021 |
| USUIR (Peng et al., 2023) | AAAI22 | 20.31 | 0.841 | - | - |
| PUGAN (Cong et al., 2023) | TIP23 | 23.05 | 0.897 | 0.608 | 2.902 |
| ADP (Zhou et al., 2023a) | IJCV23 | 22.90 | 0.892 | 0.621 | 3.005 |
| NU2Net (Guo et al., 2023) | AAAI23 | 22.38 | 0.903 | 0.587 | 2.936 |
| AST (Zhou et al., 2024) | CVPR24 | 22.19 | 0.908 | 0.602 | 2.981 |
| SMDR-IS (Zhang et al., 2024a) | AAAI24 | 23.71 | 0.922 | - | - |
| Reti-Diff Fang et al. (2025) | ICLR25 | 24.12 | 0.910 | 0.631 | 3.088 |
| **ILVS²Net** | Ours | 24.48 | 0.934 | 0.843 | 4.182 |

Table 3: Results on the BIE task.

| Methods | Sources | *BAID* | | | |
|---|---|---|---|---|---|
| | | PSNR ↑ | SSIM ↑ | LPIPS ↓ | FID ↓ |
| EnGAN (Jiang et al., 2021) | TIP21 | 17.96 | 0.819 | 0.182 | 43.55 |
| URetinex (Wu et al., 2022b) | CVPR22 | 19.08 | 0.845 | 0.206 | 42.26 |
| CLIP-LIT (Liang et al., 2023) | ICCV23 | 21.13 | 0.853 | 0.159 | 37.30 |
| Diff-Retinex Yi et al. (2023) | ICCV23 | 22.07 | 0.861 | 0.160 | 38.07 |
| DiffIR Xia et al. (2023) | ICCV23 | 21.10 | 0.835 | 0.175 | 40.35 |
| AST Zhou et al. (2024) | CVPR24 | 22.61 | 0.851 | 0.156 | 32.47 |
| MambaIR Guo et al. (2024) | ECCV24 | 23.07 | 0.874 | 0.153 | 29.13 |
| RAVE Gaintseva et al. (2024) | ECCV24 | 21.26 | 0.872 | 0.096 | 64.89 |
| Reti-Diff Fang et al. (2025) | ICLR25 | 23.19 | 0.876 | 0.147 | 27.47 |
| **ILVS²Net** | Ours | 24.89 | 0.910 | 0.085 | 31.36 |

Table 4: Results on the FIE task.

| Methods | Sources | *Fundus* | | |
|---|---|---|---|---|
| | | BIQE ↓ | CLIPIQA ↑ | FID ↓ |
| SNR-Net Xu et al. (2022) | CVPR22 | 6.144 | 0.557 | 79.284 |
| URetinex (Wu et al., 2022b) | CVPR22 | 12.158 | 0.561 | 33.347 |
| SCI Ma et al. (2022) | CVPR22 | 23.527 | 0.552 | 85.175 |
| MIRNetV2 Zamir et al. (2022b) | TPAMI22 | 14.925 | 0.527 | 47.607 |
| FourLLE Wang et al. (2023a) | MM23 | 7.741 | 0.508 | 28.736 |
| CUE Zheng et al. (2023b) | ICCV23 | 11.721 | 0.448 | 111.336 |
| Retformer Yang et al. (2023) | ICCV23 | 6.054 | 0.564 | 29.316 |
| Reti-Diff Fang et al. (2025) | ICLR25 | 10.788 | 0.525 | 27.637 |
| CIDNet Yan et al. (2025) | CVPR25 | 10.663 | 0.529 | 41.089 |
| **ILVS²Net** | Ours | 6.415 | 0.565 | 25.170 |

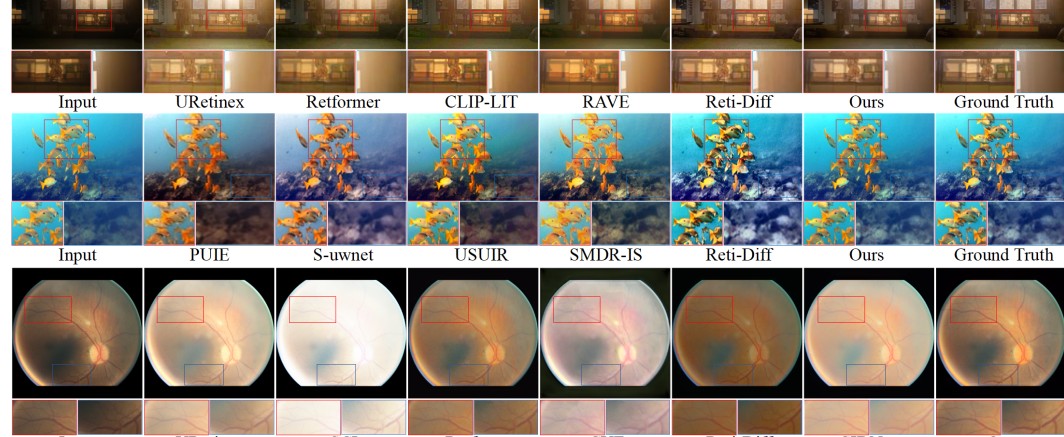

| Input | URetinex | Retformer | CLIP-LIT | RAVE | Reti-Diff | Ours | Ground Truth |
|---|---|---|---|---|---|---|---|
| Input | PUIE | S-uwnet | USUIR | SMDR-IS | Reti-Diff | Ours | Ground Truth |
| Input | URetinex | SCI | Retformer | CUE | Reti-Diff | CIDNet | Ours |

Figure 4: Visual results on the Backlit image enhancement, Underwater image enhancement and Fundus image enhancement task.

### 4.1 COMPARATIVE EVALUATION

**Low-light Image Enhancement**: Following Reti-Diff Fang et al. (2025), We evaluate our model on three benchmarks—*LOL-v1* Wei et al. (2018), *LOL-v2-real* Yang et al. (2021), and *LOL-v2-syn* Yang et al. (2021)—using four metrics: PSNR, SSIM, FID Heusel et al. (2017), and BIQE Moorthy & Bovik (2010). Superior performance is indicated by higher PSNR and SSIM scores as well as lower FID and BIQE values. Figure 3 and Table 1 demonstrate that our method achieves top performance across all three datasets, delivers superior visual quality, and maintains a compact model size, thereby underscoring its exceptional effectiveness. Additional visual results of low-light enhancement can be seen in Figure 8.

**Underwater image enhancement**. We validate our method on the *UIEB* dataset Li et al. (2019) using four widely adopted metrics—PSNR, SSIM, UCIQE (Yang & Sowmya, 2015), and UIQM (Panetta et al., 2015)—where higher scores indicate better enhancement quality. As shown in Table 2, our approach outperforms all competing methods across every metric. Furthermore, the visual examples in Figure 4 demonstrate its strong capacity to correct color distortions and significantly enhance fine textures in underwater scenes.

**Backlit image enhancement**. Following CLIP-LIT Liang et al. (2023), we trained our network on the BAID Lv et al. (2022) dataset, evaluating with PSNR, SSIM, LPIPS(Zhang et al., 2018) and FID. Table 3 shows our method consistently outperforms existing approaches, demonstrating its capability to enhance backlit images by reducing artifacts and improving quality. Figure 4 shows our model's strength in brightness enhancement while preserving detail.

**Fundus image enhancement**. Following the evaluation protocol of Reti-Diff Fang et al. (2025), we evaluate our model on the *Fundus* dataset using weights pretrained on *LOL-v2-syn*. We report BIQE, CLIPIQA Wang et al. (2023b) (higher is better), and FID. As summarized in Table 4 and illustrated in Figure 4, our method maintains a leading position in both quantitative metrics and qualitative visual results.

**Real-world illumination degradation image restoration**. We tested our method on real-world IDIR tasks using four datasets: LIME Guo et al. (2016), MEF Wang et al. (2013), NPE Ma et al. (2015), and VV He et al. (2024), following the strategy of CIDNet Feng et al. (2024). Since these images lack high-quality ground-truths, we used a pre-trained model on LOL-v2-syn and evaluated using

Figure 5: Ablation Study Breakdown with NLVSS Structure Diagram

Table 5: Results on the real-world IDIR task.

| Methods | Sources | NPE | | LIME | | MEF | | VV | |
|---|---|---|---|---|---|---|---|---|---|
| | | PI↓ | NIQE↓ | PI↓ | NIQE↓ | PI↓ | NIQE↓ | PI↓ | NIQE↓ |
| EnGAN (Jiang et al., 2021) | TIP21 | 4.015 | 4.705 | 3.669 | 4.593 | 4.015 | 4.705 | 3.386 | 4.047 |
| KinD++ (Zhang et al., 2021a) | IJCV21 | 3.785 | 4.908 | 3.785 | 4.908 | 4.016 | 4.557 | 3.773 | 3.822 |
| SNR-Net (Xu et al., 2022) | CVPR22 | 3.753 | 5.937 | 3.753 | 5.937 | 3.677 | 6.449 | 3.503 | 9.506 |
| DCC-Net (Zhang et al., 2022) | CVPR22 | 3.312 | 4.425 | 3.312 | 4.425 | 3.424 | 4.598 | 3.615 | 3.286 |
| UHDFor (Li et al., 2023) | ICLR23 | 4.124 | 4.430 | 4.124 | 4.430 | 3.813 | 4.231 | 3.319 | 4.330 |
| PairLIE (Fu et al., 2023) | CVPR23 | 3.387 | 4.587 | 3.387 | 4.587 | 4.133 | 4.065 | 3.334 | 3.574 |
| GDP (Fei et al., 2023) | CVPR23 | 4.115 | 4.891 | 4.115 | 4.891 | 3.694 | 4.609 | 3.431 | 4.683 |
| Reti-Diff Fang et al. (2025) | ICLR25 | **2.837** | 3.693 | **3.111** | **4.128** | 2.876 | 3.554 | **2.651** | **2.540** |
| CIDNet Yan et al. (2025) | CVPR25 | 2.985 | **3.550** | 3.146 | 4.132 | **2.683** | 3.568 | 2.826 | 3.218 |
| **ILVS²Net** | Ours | 2.952 | 3.381 | 3.138 | 4.093 | 2.748 | **3.377** | 2.595 | 2.317 |

Table 6: Low-light image detection on *ExDark*.

| Methods (AP) | Bicycle | Boat | Bottle | Bus | Car | Cat | Chair | Cup | Dog | Motor | People | Table | Mean |
|---|---|---|---|---|---|---|---|---|---|---|---|---|---|
| Baseline | 74.7 | 64.9 | 70.7 | 84.2 | 79.7 | 47.3 | 58.6 | 67.1 | 64.1 | 66.2 | 73.9 | 45.7 | 66.4 |
| RetinexNet | 72.8 | 66.4 | 67.3 | 87.5 | 80.6 | 52.8 | 60.0 | 67.8 | 68.5 | 69.3 | 71.3 | 46.2 | 67.5 |
| KinD | 73.2 | 67.1 | 64.6 | 86.8 | 79.5 | 58.7 | 63.4 | 67.5 | 67.4 | 62.3 | 75.5 | 51.4 | 68.1 |
| MIRNet | 74.9 | 69.7 | 68.3 | 89.7 | 77.6 | 57.8 | 56.9 | 66.4 | 69.7 | 64.6 | 74.6 | 53.4 | 68.6 |
| RUAS | 75.7 | 71.2 | 73.5 | 90.7 | 80.1 | 59.3 | 67.0 | 66.3 | 68.3 | 66.9 | 72.6 | 50.6 | 70.2 |
| Restormer | 77.0 | 71.0 | 68.8 | **91.6** | 77.1 | 62.5 | 57.3 | 68.0 | 69.6 | 69.2 | 74.6 | 49.7 | 69.7 |
| SCI | 73.4 | 68.0 | 69.5 | 86.2 | 74.5 | 63.1 | 59.5 | 61.0 | 67.3 | 63.9 | 73.2 | 47.3 | 67.2 |
| SNR-Net | 78.3 | 74.2 | 74.5 | 89.6 | **82.7** | 66.8 | 66.3 | 62.5 | 74.7 | 63.1 | 73.3 | 57.2 | 71.9 |
| Reti-Diff | **82.0** | **77.9** | 76.4 | **92.2** | 83.3 | 69.6 | 67.4 | 74.4 | 75.5 | 74.3 | **78.3** | 57.9 | 75.8 |
| **ILVS²Net** | 88.6 | 81.1 | 79.1 | 87.9 | 73.9 | 75.9 | 80.9 | 81.1 | 83.8 | 85.3 | 70.3 | **64.7** | 79.4 |

PIBlau et al. (2018) and NIQE Mittal et al. (2012), where lower scores indicate better results. As shown in Table 5 and Figure 7, our method consistently outperforms competing approaches.

## 4.2 ABLATION STUDY

In this section, we conduct ablation studies on the LOL-v2-syn dataset to assess the impact of different components of our model and the influence of the number of stages.

**Analysis of the Unrolling Stage $T$:** The choice of unrolling depth $T$ directly influences the granularity with which our network approximates the Retinex decomposition. We evaluate $T \in \{2, 3, 4, 5\}$ (see Table 7) and observe a clear trade-off: At $T = 2$, the network capacity is limited, yielding a PSNR of 25.73dB and SSIM of 0.936. Increasing to $T = 3$ boosts performance to a peak PSNR of 26.16dB and SSIM of 0.960, indicating sufficient iteration for accurate illumination–reflectance separation without overfitting. Further unrolling to $T = 4$ and 5 introduces diminishing returns and even slight performance drops (e.g., PSNR falls to 26.14dB at $T = 4$ and 26.05dB at $T = 5$), likely due to error accumulation across more steps and increased model complexity. Thus, $T = 3$ strikes the best balance between decomposition fidelity and robustness, maximizing quantitative scores while avoiding the risk of over-parameterization.

**Analysis of Core Module Contributions:** To verify the contribution of each key module in ILVS²Net—NLVSS, ISP, $P_\theta^{(R)}$, and $P_\theta^{(GS)}$—we conduct a series of ablations (Figure 5). Specifically, we replace the illumination proximal-operator network $P_\theta^{(GS)}$ (denoted $\mathcal{R}(\cdot)$) with CNN, non-local (NL), visual state space (VSS), and NL+VSS, denoted by $\mathcal{R}_1(\cdot)$, $\mathcal{R}_2(\cdot)$, $\mathcal{R}_3(\cdot)$, and $\mathcal{R}_4(\cdot)$, respectively. As reported in Table 7, substituting NLVSS with these classical modules, or removing it altogether, consistently degrades performance. Moreover, as illustrated in Figure 5, retaining NLVSS yields illumination maps that align more closely with real-world lighting. We further replace the illumination smoothing operator $\varpi(\cdot)$ (ISP) with WLS, TV, and a shallow CNN, denoted $\varpi_1(\cdot)$, $\varpi_2(\cdot)$, and $\varpi_3(\cdot)$. Table 7 shows that replacing or removing ISP also leads to performance drops. Qualitatively (Figure 5), omitting the illumination smoothing operator results in more uneven illumination, confirming its role in enforcing spatial smoothness. Finally, removing the reflectance proximal-operator network $P_\theta^{(R)}$ causes noticeable performance degradation, introduces a visible loss of fine texture, and produces artifacts in the enhanced images (Figure 5). Collectively, the quantitative drops and qualitative distortions—loss of global consistency without NLVSS, spotted/uneven illumination without ISP, and texture degradation without $P_\theta^{(R)}$—underscore the necessity of our proposed modules for achieving clear, artifact-free enhancement.

**Other configurations in ILVS²Net.** To further assess the effectiveness and generalizability of our modules, we plug NLVSS and ISP into two representative baselines—*CUE* (a deep unfolding method) and *MambaLLIE* (a Mamba-based pure neural model)—and denote the augmented variants as *MambaLLIE++* and *CUE++*, respectively. For *MambaLLIE*, we insert NLVSS and ISP immediately after concatenating the input image's channel-wise maximum and minimum features, which facilitates illumination enhancement and smoothing. For *CUE*, at each stage we apply NLVSS followed

Table 7: Ablation study in the LLIE task.

| Datasets | Metrics | Effect of NLVSS | | | | Effect of DIQM and RIFM | | | Effect of ISP | | | ILVS²Net (Ours) |
|---|---|---|---|---|---|---|---|---|---|---|---|---|
| | | $\mathcal{R}_1(\cdot)\to\mathcal{R}(\cdot)$ | $\mathcal{R}_2(\cdot)\to\mathcal{R}(\cdot)$ | $\mathcal{R}_3(\cdot)\to\mathcal{R}(\cdot)$ | $\mathcal{R}_4(\cdot)\to\mathcal{R}(\cdot)$ | w/o $P_\theta^{(GS)}$ | w/o $P_\theta^{(R)}$ | w/o $\varpi$ | $\varpi_1(\cdot)\to\varpi(\cdot)$ | $\varpi_1(\cdot)\to\varpi(\cdot)$ | $\varpi_1(\cdot)\to\varpi(\cdot)$ | |
| $L\text{-}v2\text{-}s$ | PSNR ↑ | 25.68 | 25.64 | 25.77 | 25.18 | 25.74 | 25.77 | 25.84 | 25.32 | 25.48 | 25.62 | **26.16** |
| | SSIM ↑ | 0.936 | 0.932 | 0.938 | 0.931 | 0.936 | 0.940 | 0.939 | 0.943 | 0.938 | 0.942 | **0.960** |
| $L\text{-}v2\text{-}r$ | PSNR ↑ | 22.86 | 22.82 | 22.85 | 22.80 | 22.72 | 22.74 | 22.78 | 22.83 | 22.81 | 22.80 | **23.06** |
| | SSIM ↑ | 0.882 | 0.883 | 0.886 | 0.880 | 0.883 | 0.885 | 0.883 | 0.884 | 0.882 | 0.880 | **0.887** |

Table 8: Performance of ILVS²Net with different restoration models and stage numbers.

| Datasets | Metrics | Restoration models | | | | Stage numbers | | | ILVS²Net |
|---|---|---|---|---|---|---|---|---|---|
| | | mambaille | mambaille++ | CUE | CUE++ | K=2 | K=4 | K=5 | IDIRM, K=3 |
| $L\text{-}v2\text{-}s$ | PSNR↑ | 25.87 | 25.96 | 24.41 | 25.11 | 25.73 | 26.14 | 26.05 | **26.16** |
| | SSIM↑ | 0.940 | 0.946 | 0.917 | 0.931 | 0.936 | 0.954 | 0.957 | **0.960** |
| | LPIPS↑ | 0.063 | 0.057 | 0.097 | 0.079 | 0.042 | 0.039 | 0.037 | **0.035** |

Table 9: Experiments under the unsupervised setting.

| Dataset | Metric | NeRCO Yang et al. (2023) ICCV'23 | PairLIE Liang et al. (2023) CVPR'23 | LightenDiff Jiang et al. (2024) ECCV'24 | UnfoldIR Ours |
|---|---|---|---|---|---|
| $L\text{-}v1$ | PSNR↑ | 19.84 | 19.51 | 20.45 | **21.07** |
| | SSIM↑ | 0.771 | 0.731 | 0.803 | **0.811** |

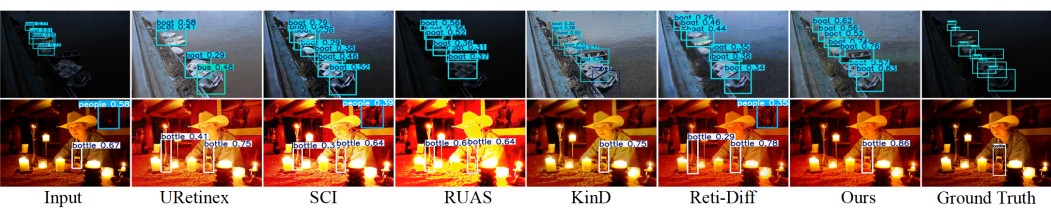

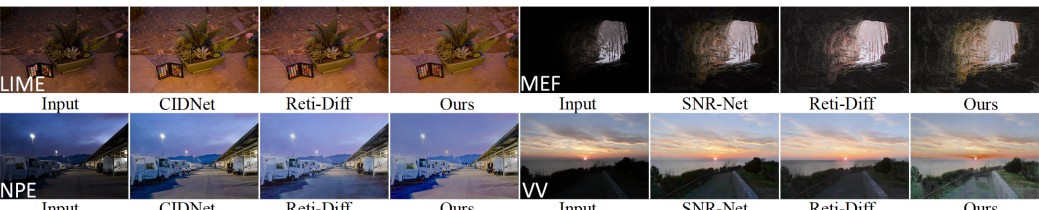

Input  URetinex  SCI  RUAS  KinD  Reti-Diff  Ours  Ground Truth

Figure 6: Visual results on the low-light object detection task.

Input  CIDNet  Reti-Diff  Ours  Input  SNR-Net  Reti-Diff  Ours

Input  CIDNet  Reti-Diff  Ours  Input  SNR-Net  Reti-Diff  Ours

Figure 7: Visual results on the real-world illumination degradation image restoration task.

by ISP on the illumination branch. As shown in Table 8, augmenting either baseline with our modules consistently improves performance, confirming that the proposed priors transfer well to other illumination-enhancement architectures.

**Potential applications of ILVS²Net:** We further explore the potential of ILVS²Net by extending it to the unsupervised setting. Following the experimental protocol of LightenDiff and incorporating the SCIMa et al. (2022) loss, our framework attains strong performance under unsupervised training. As shown in Table 9, with well-designed unsupervised losses, our method surpasses current state-of-the-art approaches, highlighting its effectiveness not only in the supervised regime but also in the unsupervised one.

### 4.3 ANALYSIS ON DOWNSTREAM TASKS

To validate whether our enhancement benefits downstream tasks, we evaluate its impact on object detection under low-light conditions. Following the setup in Fang et al. (2025), we feed the images processed by each method into a YOLO detector and test on the *ExDark* dataset Loh & Chan (2019). As shown in Table 6 and Figure 6, ILVS²Net achieves the highest detection accuracy among all compared methods, demonstrating its effectiveness in improving high-level vision performance.

## 5 CONCLUSION

We propose ILVS²Net, a Retinex-inspired deep-unfolding network that injects a group-sparse prior into every iteration. Building on a principled derivation grounded in illumination physics, we design two new proximal-operator networks—NLVSS for non-local visual state-space regularization and ISP for illumination smoothing—and integrate them as learnable priors. By preserving dynamic illumination variations and reflectance, the model mitigates information loss and color distortion across both homogeneous and textured regions. On five standard LLIE benchmarks, ILVS²Net consistently surpasses state-of-the-art methods; ablations further show that the group-sparsity-driven proximal modules provide complementary gains in spatial smoothness and structural fidelity.

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

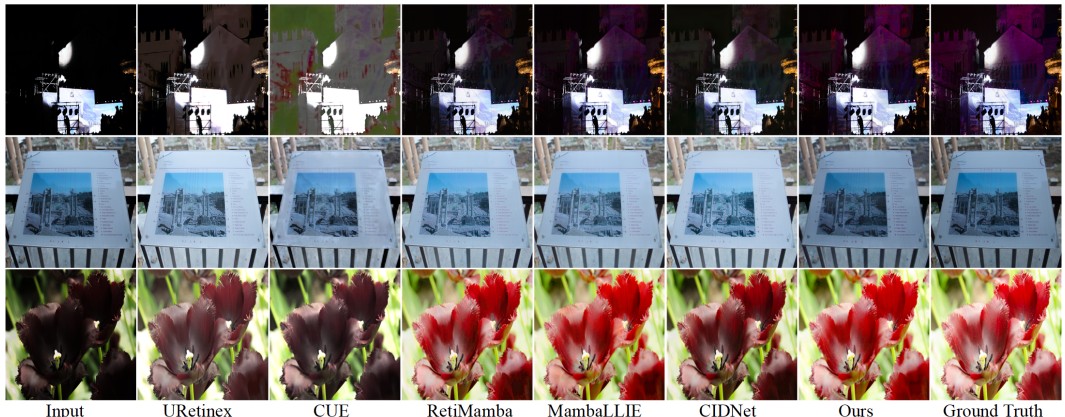

| Input | URetinex | CUE | RetiMamba | MambaLLIE | CIDNet | Ours | Ground Truth |

Figure 8: More visual results on the low-light image enhancement task.

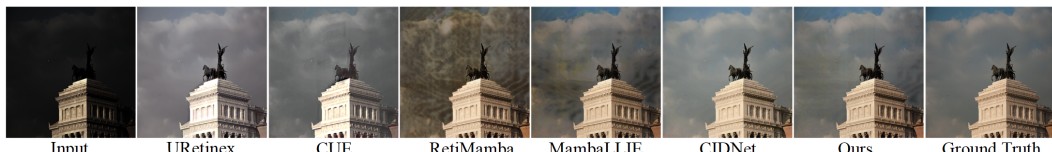

| Input | URetinex | CUE | RetiMamba | MambaLLIE | CIDNet | Ours | Ground Truth |

Figure 9: Limitations.

# A APPENDIX

## MORE VISUAL RESULTS

## LIMITATIONS AND FUTURE WORK

As shown in Fig. 9, although our method excels at maintaining smooth and consistent illumination, it still struggles to recover subtle texture details. This issue mainly stems from noise amplification during the illumination restoration process. According to Retinex theory, the reflectance component represents an object's intrinsic physical properties and should remain unchanged; however, noise infiltrates the reflectance, causing image distortion and loss of fine textures. To address this, future work will investigate the physical characteristics of the reflectance component more deeply, with the goal of effectively suppressing noise in the reflectance during restoration and thereby improving detail preservation.

