# OpenReview forum: "ILVS\(^2\)Net: Illumination-Driven Non-Local Visual State Space Unfolding Network for Low-Light Enhancement"
_ICLR.cc/2026/Conference — ICLR 2026 Conference Withdrawn Submission_

### Official Review · Reviewer_ZpMx · 2025-10-30

**Soundness:** 2
**Presentation:** 3
**Contribution:** 2
**Rating:** 2
**Confidence:** 5

**Summary:**

This paper presents ILVS2Net, a deep unfolding network for low-light image enhancement (LLIE) that attempts to integrate a group-sparse prior into the Retinex-based deep unfolding process. The authors introduce two new modules: the Non-Local Visual State Space (NLVSS) and the Illumination Smoothing Operator (ISP), which are designed to address issues such as oversmoothing and instability in existing methods. Despite its technical depth, the paper does not offer a significant innovation in the field. While the approach improves some existing methods, the contributions appear incremental rather than groundbreaking. The experiments, although extensive, do not sufficiently demonstrate a substantial leap over the current state-of-the-art methods in LLIE.

**Strengths:**

1. The paper is well-structured and easy to follow. The authors take the time to explain the methodology clearly, breaking down complex concepts in a way that is accessible.
2. Despite being a relatively complex model, ILVS2Net is designed efficiently in terms of both computation and memory usage.
3. While traditional methods in low-light image enhancement typically rely on pixel-level or patch-level regularization, the group-sparse prior in ILVS2Net models illumination as a non-local group of structurally similar patches.

**Weaknesses:**

1. One of the main weaknesses of this paper is the lack of novelty. While the integration of non-local dependencies and illumination smoothing is a reasonable approach, it doesn’t represent a breakthrough in the field of low-light image enhancement. Many recent methods already explore similar ideas, such as Retinex-based models, non-local operations, and group sparsity priors. The contribution here feels incremental, and the authors do not convincingly argue why their specific implementation is a significant improvement over existing methods like MambaIR, RetinexFormer, or others.
2. While the technical design of the network and modules is well-executed, the paper spends more time detailing the architecture and optimization methods than exploring new, groundbreaking concepts. There is a lack of a well-defined, new theoretical framework or insight.
3. The paper does not provide a clear theoretical explanation of why these specific modules work well or why they are necessary for improving low-light image enhancement. There is little discussion on the theoretical foundations of the group-sparse prior or the non-local modeling approach in the context of Retinex decomposition.

**Questions:**

Please list clearly any key questions or suggestions for the authors.
1.How do the proposed modules compare with other recent methods that incorporate similar priors, such as group sparsity or non-local modeling?
2.Could the authors provide more details on how the model's structure can be adapted or extended to other image restoration tasks?

---

### Official Review · Reviewer_y4Ks · 2025-10-31

**Soundness:** 3
**Presentation:** 2
**Contribution:** 2
**Rating:** 4
**Confidence:** 4

**Summary:**

This manuscript performs low-light enhancement based on Retinex's unfolded network to solve the shortcomings of current methods for illumination estimation. Starting from the sparsity of illumination structure, two learnable operator networks are introduced to optimize the estimation of illumination and reflectance, and alleviate the problem of information loss and color distortion. Experiments show the effectiveness of the method for downstream tasks.

**Strengths:**

For illumination estimation in a deep Retinex’s unfolded network, non-local visual state space is introduced, and a learnable operator network is used to obtain a group sparse prior that captures global information. Secondly, another network is introduced to dynamically adjust the estimation of illumination.

**Weaknesses:**

1. The technical details described in the manuscript are not clear enough, which affects the reviewer's understanding of the technical innovation of the paper. Below are some examples.
  - What is the structured sparsity of illumination？
  - What do “i“ and ”*“ in Eq. 2 mean, respectively?
  - How are similar patches or channels obtained in 262-263?
  - Why is the input of the IEM in Figure 2 inconsistent with the input of the detailed figure? There seems to be confusion?
  - Why should the input of RVSS also contain L, shouldn't it be input only R? What happens if you just input R?
  - Many capitalized proper nouns in the paper do not specify the detailed meaning and may cause confusion.
2. The consistency between the charts and data in the experimental verification phase is poor.
  - Table 7 Presence of symbol representation errors: Effect of ISP. Table 8 There are indicator symbol errors
  - Why is the result of removing the reflectance proximal-operator network poorly represented in Figure 5, but the metrics are relatively high in Table 7?
  - The results in Figure 7 show that there seem to be more blocky and green artifacts. Why does this happen?

**Questions:**

1. In 288-302, ISP also need to be learned? As the core contribution, I think it needs to be marked more clearly in the figure of framework.
2. Figure 1 shows that other methods have limited receptive fields, is a CNN+transformer type framework considered?
3. Are there any visualization examples of the decomposed R and L?

---

### Official Review · Reviewer_DXRV · 2025-10-31

**Soundness:** 2
**Presentation:** 3
**Contribution:** 3
**Rating:** 4
**Confidence:** 4

**Summary:**

This paper proposes a deep unrolled network (ILVS2NET) for low-light image enhancement. This method is based on the Retinex decomposition model and regularizes the restoration of the illumination map by introducing a Group-Sparse Prior in the optimization iterations. The network architecture combines a Visual State Space model with a Non-Local block to enhance feature extraction.

**Strengths:**

This paper's approach of combining a deep unrolled network with a physical model has a sound theoretical foundation and interpretability, making it a promising approach. Furthermore, the introduction of a group sparsity prior to regularize the structural information of the illumination map is an intuitive and promising improvement direction, attempting to go beyond the traditional $\ell_1$ sparsity and $\text{TV}$ smoothness. Finally, the integration of the VSS model with non-local blocks to enhance global dependency modeling demonstrates a focus on modern, efficient architectures.

**Weaknesses:**

1.The manuscript suffers from a deficit of theoretical justification and overall narrative vagueness regarding its novel components. Specifically, the paper fails to provide a rigorous theoretical or empirical explanation of how the proposed learnable modules map onto specific steps within the underlying optimization process, such as the Proximal Mapping ($\text{Proximal Map}$). Furthermore, the NLVSS module is merely embedded as a feature extractor. The theoretical correspondence linking this module to a specific operation within the group-sparse optimization iterations remains ambiguously described, suggesting a lack of deep integration beyond a superficial architectural stacking.

2.The Group-Sparse Prior is not a novel concept within the field of sparse signal processing. The authors must explicitly demonstrate how the specific formulation of their proposed Group-Sparse Prior is further refined and tailored for the characteristics of the low-light illumination component ($L$). It is essential to clarify whether this is simply a convenient substitution for the existing $\ell_1$ prior or if it represents a deeper, principled modeling of the illumination map's structured sparsity based on physical or statistical observations. If it is merely a drop-in replacement, the claimed originality is severely diminished.

3. The manuscript suffers from numerous inaccuracies in details, such as the lack of specific context for the addition in Figure 2 and the small size of the figure legend, and the lack of numbering in the formulas in the main text.

**Questions:**

1. Please explain the priors of group sparsity methods in detail, such as how groups are defined and what physical basis supports the sparsity of illumination in this particular grouping.

2. For NLVSS, please provide theoretical or empirical evidence demonstrating why the VSS module outperforms standard CNNs/Transformers in recovering illumination components during optimization, rather than simply serving as a general-purpose feature extractor.

---

### Official Review · Reviewer_8d8R · 2025-11-07

**Soundness:** 3
**Presentation:** 3
**Contribution:** 2
**Rating:** 4
**Confidence:** 4

**Summary:**

This paper presents ILVS2Net, a deep-unfolding network designed for low-light image enhancement. The authors address the challenges posed by structured sparsity in illumination, which results in oversmoothing and unstable recovery in existing methods. The proposed network integrates a group-sparse prior into each unfolding iteration by using two key components:

1) Non-Local Visual State Space (NLVSS), which captures long-range structural dependencies.

2) Illumination Smoothing Operator (ISP), which ensures edge-preserving piecewise smoothness during illumination estimation.

These components allow the network to effectively preserve dynamic illumination variations and reflectance, yielding stable and accurate results. Extensive experiments demonstrate that ILVS2Net consistently outperforms state-of-the-art methods in both quantitative metrics and perceptual quality.

**Strengths:**

1. This paper proposes ILVS2Net, a novel deep-unfolding network designed for low-light image enhancement.

2. The paper is clearly written and well-organized. The contributions are explained in sufficient detail, and the experimental results are well-supported with relevant metrics and visual examples.

3. The effectiveness of the proposed method has been demonstrated through extensive experiments.

**Weaknesses:**

1. The authors mention the structured sparsity of illumination, but do not fully explain its meaning or theoretical basis. Although they cite Kajiya (1986), the rendering equation in that paper primarily focuses on light propagation in image formation, which does not directly relate to the illumination sparsity they discuss. The concept of sparsity emphasizes structural consistency in illumination, which differs from the rendering equation. Additionally, the authors have not established a clear theoretical link between illumination sparsity, Retinex theory, and deep unfolding methods, leaving the approach insufficiently supported by theory.

2. While the authors demonstrate improvements through extensive experiments, they do not sufficiently analyze the impact of group sparse on illumination sparsity. Further theoretical and experimental analysis is needed to explain how the group sparse influence illumination sparsity and why this is effective. This would better support the significance and novelty of the proposed method.

3. The contributions of the authors are incremental. First, the authors primarily compare their method to MambaLLIE but do not make a direct contribution to the state-space model. Secondly, the Group Sparse Representation they propose is not novel, as it has already been explored [ref-1]. Finally, the use of an unfolding network structure is a well-established approach for LLIE, as seen in methods like URetinex. The authors seem to have effectively combined these existing techniques, but the overall contribution does not present groundbreaking innovations.

4. Figure 1 is not clear enough. I suggest using PDF format for better clarity and more defined structure. Additionally,the input and output are not clearly defined as "input (low-light image)" and "output (enhanced image)".

5. The paper lacks sufficient comparisons with latest unfolding network-related methods. While the results show that the method outperforms several benchmarks, it would be beneficial to compare the approach with more recent deep unfolding models.

6. On lines 184-185 and 259-260, there is a missing equation reference indexes.




[ref-1] Low-Rankness Guided Group Sparse Representation for Image Restoration. IEEE Transactions on Neural Networks and Learning Systems, 34(10), 7593-7607, 2021.

**Questions:**

please see weaknesses.

---

### Note · Authors · 2025-12-07

I have read and agree with the venue's withdrawal policy on behalf of myself and my co-authors.